# Construction of a web-based nanomaterial database by big data curation and modeling friendly nanostructure annotations

Xiliang Yan [1,2], Alexander Sedykh[2,3], Wenyi Wang[2], Bing Yan [1,4✉] & Hao Zhu [2,5✉]

Modern nanotechnology research has generated numerous experimental data for various nanomaterials. However, the few nanomaterial databases available are not suitable for modeling studies due to the way they are curated. Here, we report the construction of a large nanomaterial database containing annotated nanostructures suited for modeling research. The database, which is publicly available through http://www.pubvinas.com/, contains 705 unique nanomaterials covering 11 material types. Each nanomaterial has up to six physicochemical properties and/or bioactivities, resulting in more than ten endpoints in the database. All the nanostructures are annotated and transformed into protein data bank files, which are downloadable by researchers worldwide. Furthermore, the nanostructure annotation procedure generates 2142 nanodescriptors for all nanomaterials for machine learning purposes, which are also available through the portal. This database provides a public resource for data-driven nanoinformatics modeling research aimed at rational nanomaterial design and other areas of modern computational nanotechnology.

[1] Institute of Environmental Research at Greater Bay, Key Laboratory for Water Quality and Conservation of the Pearl River Delta, Ministry of Education, Guangzhou University, Guangzhou 510006, China. [2] The Rutgers Center for Computational and Integrative Biology, Camden, NJ 08102, USA. [3] Sciome, Research Triangle Park, North Carolina 27709, USA. [4] School of Environmental Science and Engineering, Shandong University, Jinan 250100, China. [5] Department of Chemistry, Rutgers University, Camden, NJ 08102, USA. ✉email: drbingyan@yahoo.com; hao.zhu99@rutgers.edu

The global market value of nanotechnology is expected to reach $90.5 billion by 2021[1] as commercial and consumer nano-products continue to rise[2–4]. Increased production, use and environmental accumulation of these nanomaterials present important toxicology concerns[5–7]. A variety of in vitro and in vivo assays evaluating their potential environmental and human health effects have generated vast quantities of experimental data[8,9], requiring data extraction, analysis, and sharing for guiding the safe design of next-generation nanomaterials[10,11]. This urgency is echoed in the recent Nanoinformatics Roadmap 2030 in USA and Europe, aimed at promoting the capture, preservation, and dissemination of publicly available data on nanomaterials. The Roadmap, which outlined the importance of coordinating research efforts and charting the challenges in nanoinformatics as a set of milestones, envisages the flow of data from experimentalists into structured databases that can be used by computational modelers to predict nanomaterial properties, exposure and hazard values that will support regulatory actions[12].

Two large databases for chemicals and proteins have already impacted different areas of science. As a small molecule database, PubChem provides structural annotation (e.g., chemical structures, SMILES, and InChi key), physicochemical properties (e.g., logP and molecular weight) and available bioactivities (e.g., EC50 and IC50)[13]. Since its launch in 2004, PubChem has served various scientific communities including cheminformatics, chemical biology, medicinal chemistry, and drug discovery. Another crucial database for scientific community is the Protein Data Bank (PDB)[14], which provides three-dimensional structures of biological macromolecules, (e.g., proteins and nucleic acids) as PDB files for broad researchers in fields like molecular biology, structural biology, and computational biology. However, a comparable nanomaterial database is not available. The key to building such a database of nanomaterials is nanostructure annotation—a computer-friendly format for encoding information.

Several nanomaterial databases serving specific areas are available (Table 1)[15–19]. For example, the cancer Nanotechnology Laboratory (caNanoLab) database (https://cananolab.nci.nih.gov/) built by the National Cancer Institute in 2007[15] is designed to expedite and validate the use of nanotechnology in biomedicine. However, it is not fully accessible to the public because it contains proprietary data. While these nanomaterial databases, which are shown in Table 1, share published data and have been used for modeling studies[16,20,21], they are limited by the way they are curated. Although, new file formats (e.g., JSON[17] and ISA-TAB-Nano[22]) are also specially designed in several nanomaterial databases, such as eNanomapper and NANoREG, to store and manage the curated nanomaterial data. Nanomaterial entities (e.g., composition, physicochemical properties, and biological activities of the nanomaterials) in these databases exist as text outputs extracted directly from publications, ignoring nanostructure annotations that are critical for modeling studies. As a result, variables (e.g., physicochemical properties) used in previous modeling studies were mostly experimentally generated. Without nanostructure annotations, diverse structural information for predictive modeling and other research such as nanostructure analysis and visualization cannot be performed.

Here, we report a publicly available nanomaterial database that contains annotated nanostructures of diverse nanomaterials suitable for immediate modeling research. The database, constructed from thousands of scientific papers, currently contains 705 unique nanomaterials, 1365 physicochemical property (e.g., logP, zeta potential, and hydrodynamic diameter) and 2386 bioactivity (e.g., cell viability, cellular uptake, and ROS) data points. All experimentally obtained information on the structure of the nanomaterials, such as form, size, shape, and surface ligand were annotated and stored as PDB files, which are downloadable from the web portal (http://www.pubvinas.com/). The PDB files can be used to generate nanodescriptors, which were created in-house to quantitatively represent nanostructure diversity. Using these nanodescriptors, we developed predictive models for three critical property/bioactivity endpoints of various nanomaterials using machine learning ($k$-nearest neighbor) and deep learning (deep neural network) approaches. This is the largest and the only nanomaterial database that contains nanostructure annotations to support nanomaterial modeling and rational nanomaterial design. Furthermore, the predictive models developed from this database can be used to predict three critical properties and

**Table 1 Nanomaterial databases.**

| Database | Data points | Usage | Reference |
|---|---|---|---|
| caNanoLab https://cananolab.nci.nih.gov/ | 1308 | Expedite and validate the use of nanotechnology in biomedicine | 15 |
| S²NANO http://portal.s2nano.org/ | 6854 | Develop and commercialize safe and sustainable nano-products | 16 |
| eNanomapper http://www.enanomapper.net/ | 5528 | Develop a computational framework for nanotoxicity data management | 17 |
| Nanomaterial registry http://nanohub.org/ | 2031 | Help understanding the fundamental properties of nanomaterials | 18 |
| Nanoparticle information library http://nanoparticlelibrary.net/ | 88 | Capture the information about nanomaterial physicochemical characteristics | 19 |
| NanoMILE https://ssl.biomax.de/nanomile/cgi/login_bioxm_portal.cgi | 120 | Contain characterization data and high throughput screening toxicity data of nanomaterials | — |
| DaNa Knowledge Base https://www.nanopartikel.info/en/ | _ | Help understanding the impacts of nanomaterials for humans and the environment | — |
| NanoDatabank http://nanoinfo.org/nanodatabank/ | >1000 | Design with simplicity of nanomaterial data storing and sharing | — |
| NBI Knowledgebase http://nbi.oregonstate.edu/ | 200 | Help understanding the mechanism of nanomaterial exposure effects in biological systems | — |
| Nanowerk https://www.nanowerk.com/ | 4000 | Help the nanotechnology community to research nanomaterials | — |

The low curation of existing nanomaterials's databases is limiting their application in modeling studies. Here the authors report a publicly available nanomaterial database that contains annotated nanostructures of diverse nanomaterials immediately available for modeling research studies.

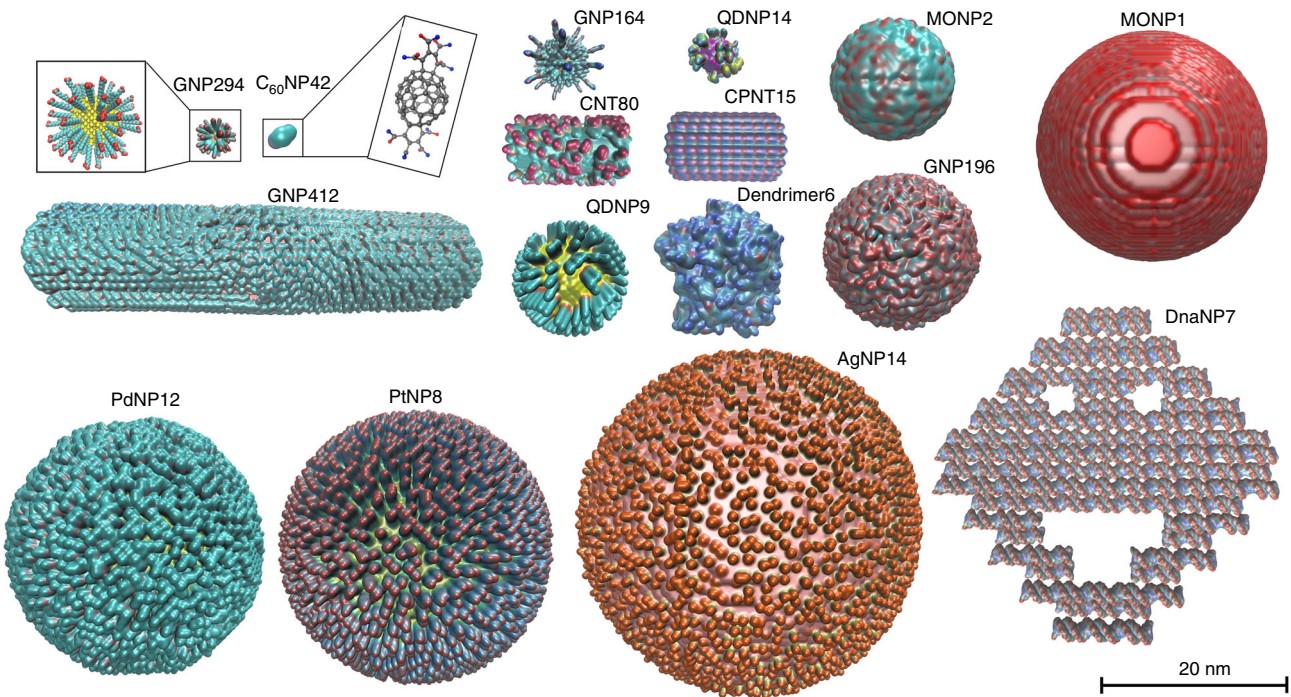

**Fig. 1 Visualization of 16 representative nanomaterials in the database.** The database contains 705 nanomaterials that vary in material type, size, shape, and surface ligand. Most nanomaterials were spherical but rod-like and irregular ones were also annotated and included in the database. Different surface chemistries of the nanostructures were rendered in different colors by QuickSurf drawing method in VMD, offering direct impressions of the nanomaterials. Emboldened text represents text identifiers that can be used to search for the nanomaterial in the database.

bioactivity (i.e., logP, zeta potentials, and cellular uptake) of new nanomaterials.

## Results

**Construction of the nanomaterial database.** A total of 705 nanomaterials, comprising 414 gold nanoparticles (GNPs), 17 silver nanoparticles (AgNPs), 12 platinum nanoparticles (PtNPs), 12 palladium nanoparticles (PdNPs), 80 carbon nanotubes (CNTs), 48 buckminsterfullerenes ($C_{60}$), 34 quantum dots (QDs), 32 metal oxides nanoparticles (MONPs), 21 DNA origami nanoparticles (DnaNPs), 11 dendrimers, and 24 cyclic peptide nanotubes (CPNTs), were annotated for the database. Figure 1 shows 16 representative nanostructures covering all nanomaterial types in the database and are rendered by visual molecular dynamics (VMD) using the QuickSurf method[23]. This method uses positions of atoms and the Monte Carlo simulation for generating the volumetric density maps and isosurface that simulate electron density and solvent accessible surface for the input nanostructures. For example, GNP164 represents the 164th gold nanoparticle in the database that has a core diameter of 5 nm (Fig. 1, see Supplementary Data for other structure information). The nanostructures varied in material type, size, shape, and surface ligand. For example, $C_{60}$NP42 and AgNP14 are 1 nm and 40 nm, respectively. Although most nanomaterials are spherical, the database also contains rod-like (e.g., GNP412, CNT80, and CPNT15) and irregular (e.g., Dendrimer6 and DnaNP7) nanomaterials. Different surface chemistries of the nanostructures were rendered with different colors. For example, the nanoparticle PdNP12 (logP = 2.52) with hydrophobic surface ligands are shown as cyan while the nanoparticle PtNP8 (logP = −1.47) with hydrophilic surface ligands are rendered purple. Other structural details can also be observed, for example, the long surface ligand chains on GNP164 are shown as tentacles. These detailed 3D plots of nanomaterials in the database provide direct

impressions of the relevant surface chemistry and physicochemical properties.

Figure 2 is an overview of the data curated in this study (see Supplementary Data for details), including physicochemical properties (logP and zeta potential), bioactivities (cell viability, reactive oxidative stress (ROS), and cellular uptake), along with the nanomaterial types and structure information (surface ligands and size). Although majority of the nanomaterials are GNPs, there are 291 other types of nanomaterials (Fig. 2a). The functions of nanomaterials are affected by surface small molecules (e.g., drugs and peptides), which determine their diverse applications (e.g., drug delivery and tumor diagnosis). As shown in Fig. 2b, the number of surface ligands ranged from 1 (such as $C_{60}$ nanomaterials) to more than 6000 (such as GNP12). This is because ligand density is highly affected by the properties of the surface ligands. For example, similar sized GNP (~5.8 nm) can have around 200 ligands per particle for positively charged ligands (e.g., GNP130) and negatively charged ligands (e.g., GNP138). Meanwhile, ligands without charges can pack up to over 700 surface ligands per GNP (e.g., GNP152). Among the 705 nanomaterials, one contained up to four different ligands (GNP392) and there were in total 314 unique surface ligands. The spherical nanomaterials in the database also had a wide size distribution (Fig. 2c). At the lower end, there are GNPs with diameter less than 10 nm that are suitable for biomedical applications[24,25]. Some spherical nanoparticles have sizes ranging from 10 to 45 nm.

The nanomaterials in this database are also biologically diverse (Fig. 2d–h). The logP values of the nanomaterials, which describe the hydrophobicity of relevant nanomaterials, ranged from −2.68 to 2.72. Zeta potential—the charge at the interface between the nanomaterial surface and its liquid medium—of nanomaterials in this database was tested in three solutions (water, aqueous buffer, and serum) and they ranged from −93.73 mV to 86.80 mV (Fig. 2e). Cell viability showed a spread from 2% to 118.05%

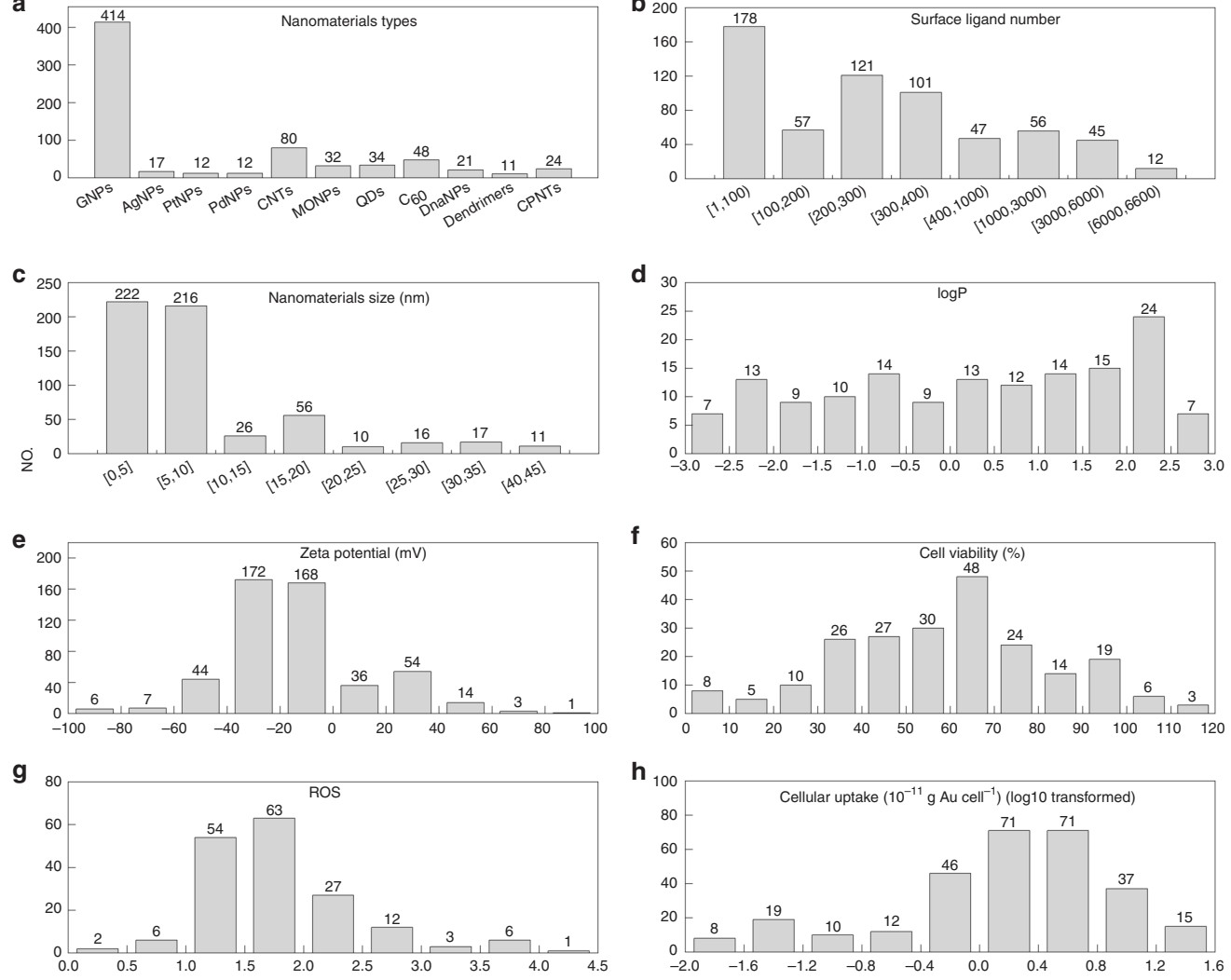

**Fig. 2 Overview of the nanomaterial database. a–h** Distributions of nanomaterials accounting to **a** nanomaterial type, **b** surface ligand number, **c** nanomaterial size, **d** logP, **e** zeta potential, **f** cell viability, **g** reactive oxidative stress (ROS) and **h** cellular uptake. Nanomaterials in the database show chemical, structural, and biological diversity. The numbers in the brackets of **b**, **c** represent the range of the surface ligand number and nanomaterial size, respectively.

(Fig. 2f), indicating the various nanomaterials induced varying degrees of cytotoxicity. ROS level, which is used to evaluate cellular oxidative stress, linked to cancer, diabetes, and aging, also ranged widely from 0.44 to 4.10 (Fig. 2g). For nanomaterials, cellular uptake is usually a prerequisite for their applications in drug delivery, bioimaging and, etc[26]. In this database, cellular uptake capacity of all nanomaterials varied from $-1.87 \, g \, cell^{-1}$ to $1.36 \, g \, cell^{-1}$ with a log10-tranformation (Fig. 2h).

**Analysis of nanostructure diversity**. After annotating and saving the structures of all 705 nanomaterials in our database as PDB files, we calculated 680 nanodescriptors using the Virtual Nanostructure Simulations (VINAS) toolbox[27]—an in-house cheminformatics program designed to calculate descriptors based on the annotated nanomaterial structures. The current descriptors calculated by VINAS are based on Delaunay tessellation, which is a fast way to transform the nano surface geometry into quantitative values as nanodescriptors. Using the 680 calculated nanodescriptors, we performed principal component analysis (PCA) and used the top three principal components, which account for 79% of the total descriptor variance, to show the occupation of all nanomaterials in a 3D chemical space

(Fig. 3a). All the nanomaterials were structurally diverse and occupied most of this chemical space. Compared to other nanomaterials, MONPs occupied a larger area because the relevant VINAS nanodescriptor values, which are based on atomic properties, varied significantly according to the unique atoms (e.g., Zn, Co, and Ce) that make up each MONPs.

Chemical structure is the key to determine a molecule's physicochemical properties and biological activities. The content that structurally similar molecules should exhibit similar bioactivities is the fundamental hypothesis of all quantitative structure-activity relationship (QSAR) and other relevant modeling studies[28,29]. To quantitatively study the structural similarity among nanomaterials, we calculated the pairwise Euclidean distance for all nanomaterials. All nanodescriptor results were normalized to a range between 0 and 1 before calculation. A total of 248,160 distances were generated among each two of the 705 nanomaterials. The distribution of values ranged from 0.004 to 17.31 with an average of 5.3 (Fig. 3b). Two substances are typically considered similar if their normalized Euclidean distance is less than 0.5[30,31]. In this database, some nanomaterials that belong to different nanomaterial types, are also structurally similar. For example, the Euclidean distances between PtNP1 and

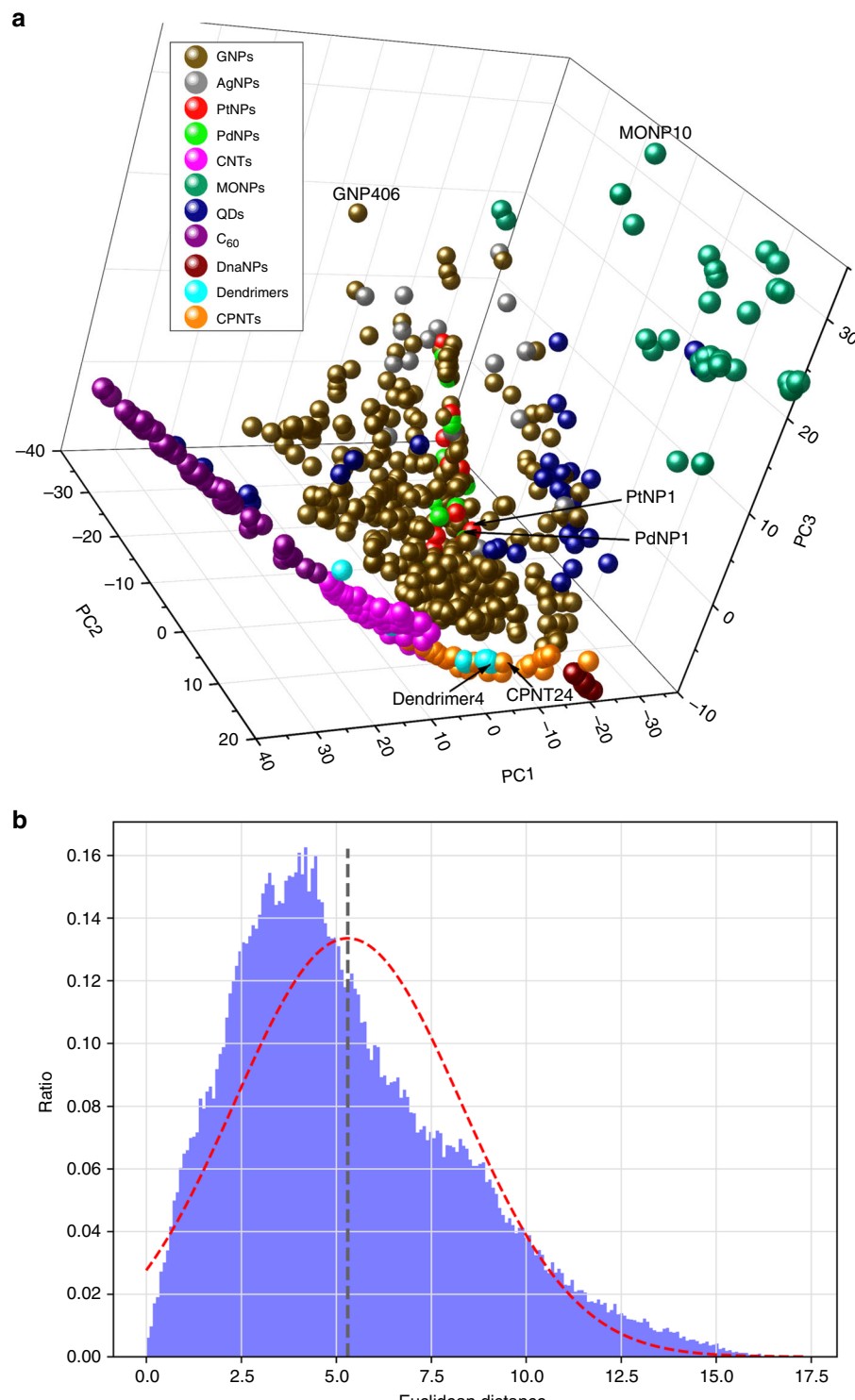

**Fig. 3 Nanostructure diversity analysis. a** Nanomaterial chemical space shown by principal component analysis (PCA) of all 705 nanomaterials. The three principal components (PC1, PC2, PC3) account for 43%, 23, and 13% of the total descriptor variance, respectively. Different colors were associated with different nanomaterial classifications. Six nanomaterials are shown with their identifiers (i.e., PtNP1, PdNP1, Dendrimer4, CPNT24, GNP406, and MONP10). **b** Distribution of the 248, 160 Euclidean distances calculated from each pair of nanomaterials in the database. The distribution ranged from 0.004 to 17.31with an average of 5.3 (black dashed line). Normalized distribution curve is shown as red dotted line.

PdNP1, and between Dendrimer4 and CPNT24 are 0.037 and 0.14, respectively. PtNP1 and PdNP1 with Euclidean distance near zero are considered structurally similar because they are about the same size (6 nm and 5.8 nm, respectively) and have the same surface ligand at the similar density (371 and 365 ligands per particle, respectively). Although Dendrimer4 is irregular and CPNT24 is rod-like, they are considered structurally similar because they have similar sizes (2 nm and 1.41 nm * 1.44 nm) and atomic compositions (C, N, O, and H). Some structural outliers such as GNP406 and MONP10 were also seen. GNP406 is

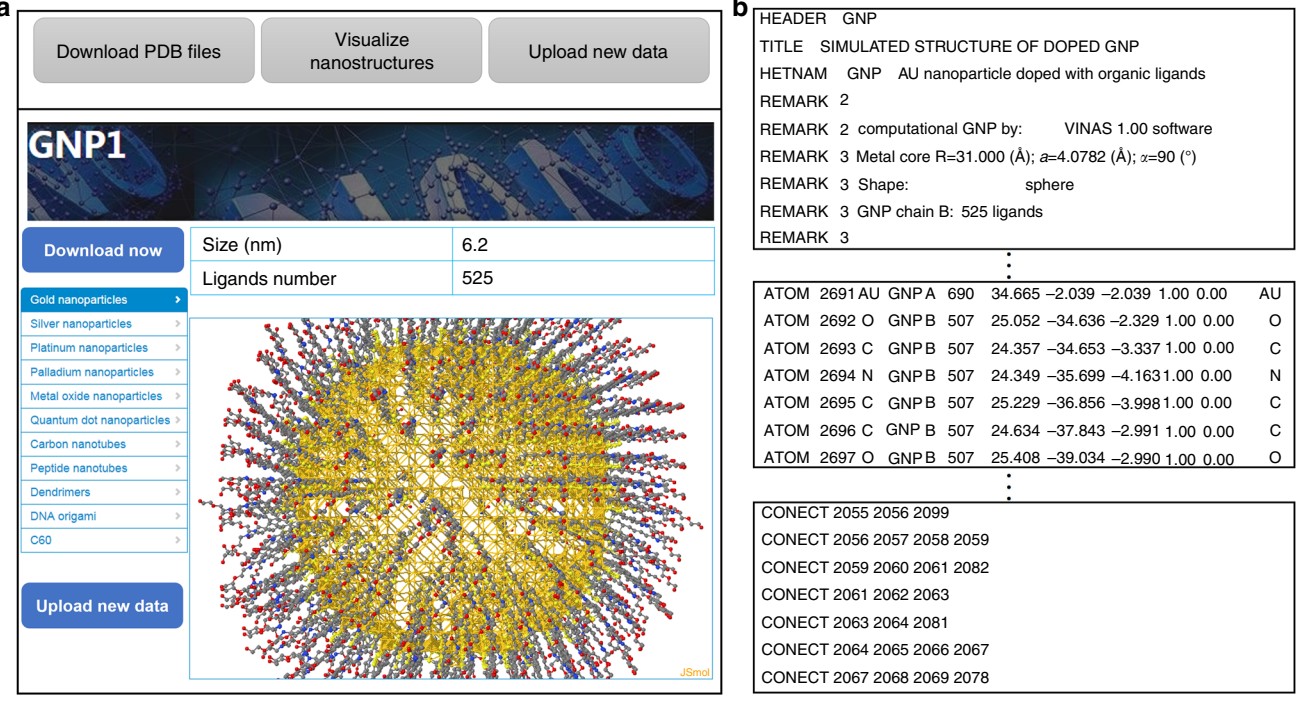

**Fig. 4 PubVINAS online portal. a** Screenshot of PubVINAS. The bars on the above and left of the picture show the user functions (e.g., download/upload data, visualize data, select data based on classifications, and, etc.) and **b** example PDB file as an output shown as three parts: (1) the basic information, (2) atom type and coordinates, and (3) the connections between atoms.

structurally different because it is a rod-like gold nanoparticle (most are spherical) that is also relatively large at 30 nm × 33 nm. MONP10, which is a $La_2O_3$ metal oxide nanoparticle around 24.6 nm in diameter, is structurally different because of the unique properties of the Lanthanum (La) atom.

**Nanomaterial database portal**. To share the structural annotated data, we developed an online database portal (http://www.pubvinas.com/) that currently can be used to download the PDB files, visualize the nanostructures and upload new data (Fig. 4a). A full-time computer systems administrator will be responsible for maintaining the portal. Each PDB file of the nanomaterials can be downloaded by clicking the dropdown bars with their corresponding classification (e.g., gold nanoparticles, silver nanoparticles, and platinum nanoparticles). Users can view the nanostructure online from the corresponding PDB file and open the downloaded PDB file using well-known cheminformatics software (e.g., VMD, RasMol, and MOE). An example PDB file is shown in Fig. 4b. The first part of the file contains the basic information on the structure of the nanomaterial (e.g., the form, shape and size); the second part contains information about the atoms (e.g., atom type and coordinates); and the third part includes information on the bond/connection between atoms. Users may also share their new data (e.g., new nanomaterials synthesized and/or tested against new bioassays) by uploading them as a text file (Fig. 4a). After reviewing the upload files, the system administrator will generate the PDB files and add the new dataset to the nanomaterial database. We expect to add more functions, such as an online toolbox to calculate nanodescriptors and several trained models, in the future to predict the properties of new nanomaterials.

**Predictive nano property/bioactivity modeling**. Using data from the database, we used $k$-Nearest Neighbor ($k$NN), a traditional machine learning approach, and deep neural network (DNN), a

representative deep learning algorithm, to build computational models that will identify quantitative relationships between the annotated nanostructures and target activities. Two properties and one bioactivity (i.e., logP, zeta potential tested in water at pH = 7, and cellular uptake capacity in A549 cells) were selected for modeling. The logP dataset contains 147 unique nanomaterials, including 123 GNPs, 12 PtNPs and 12 PdNPs. The zeta potential dataset contains 213 unique nanomaterials, including 148 GNPs, 6 AgNPs, 12 PtNPs, 12 PdNPs, 8 MONPs, 24 QDNPs, and 3 Dendrimers. The cellular uptake dataset contains 71 GNPs, which were tested against A549 cells. Each model was developed using the $k$NN and DNN approach with VINAS nanodescriptors calculated from the associated nanomaterials in the dataset. The performance of the model was evaluated by both the 5-fold cross-validation and external prediction methods common in modeling studies[32,33]. For each endpoint, the available data were randomly split into a training set (80% of the data) for developing the model, and a test set (20% of the data) for external validation of the model. The training set was further split into five subsets. The model was developed using four of the five subsets and the remaining subset was used for validation. This procedure was repeated five times until all subsets were used for validation once.

The correlations between experimental and predicted values of the six resulting models based on $k$NN and DNN are shown in Fig. 5, which also includes the root mean square error (RMSE) and correlation coefficients ($R^2$). Overall, both $R^2$ and RMSE for 5-fold cross validation ($R^2$_5CV and RMSE_5CV) and external prediction ($R^2$_val and RMSE_val) are at the same order of magnitude, indicating the 5-fold cross-validation process and external prediction yielded similar results. All correlation coefficients (both $R^2$_5CV and $R^2$_val) were above 0.5, indicating that all six models successfully predicted the relationships between the annotated the nanostructures and target activities[34]. When comparing $R^2$_5CV and $R^2$_val, $k$NN models (Fig. 5a, c, e) showed better predictability than DNN models (Fig. 5b, d, f). Although DNN is a popular modeling tool and has demonstrated

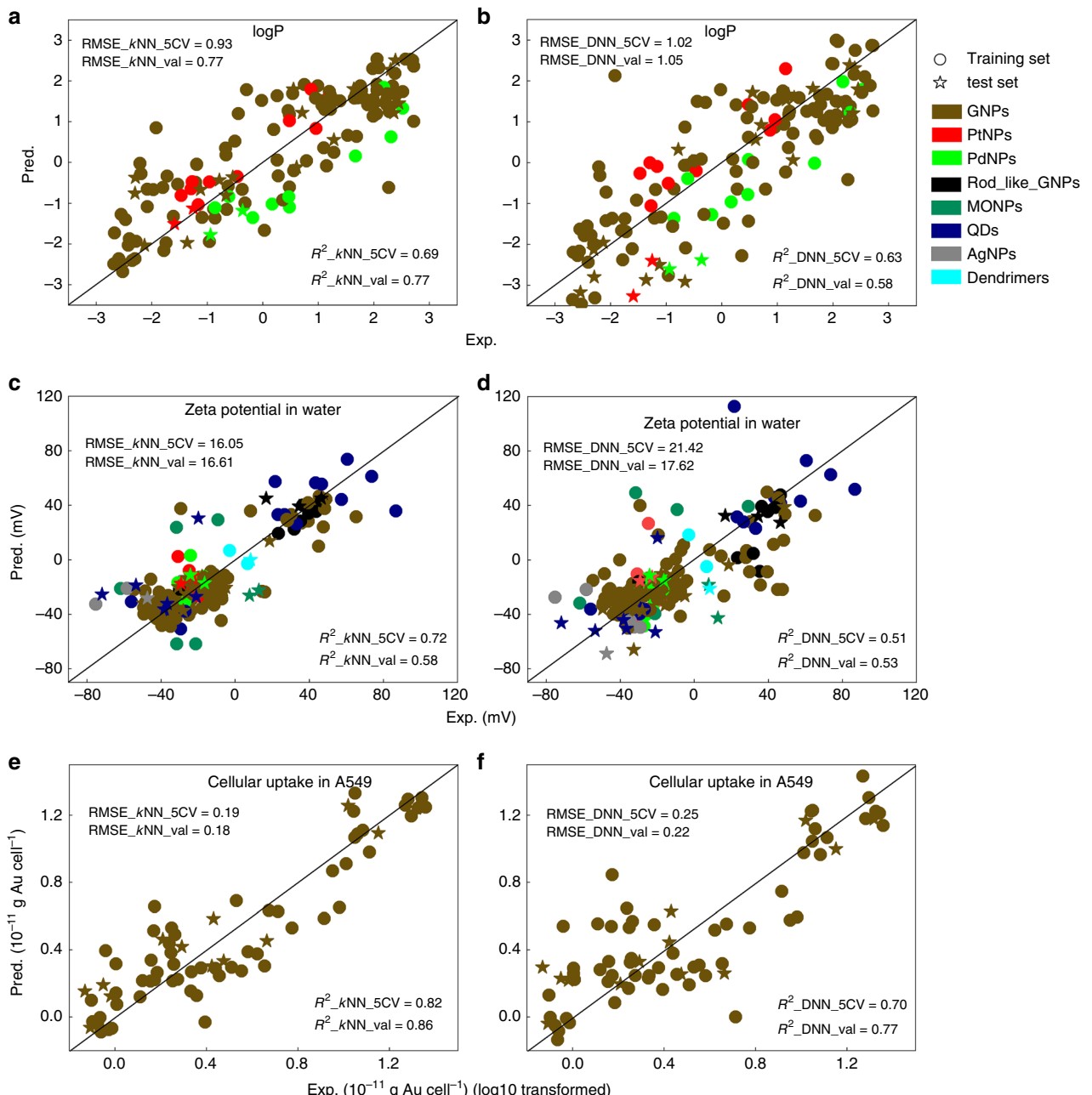

**Fig. 5 Correlations between experimental (Exp) and predicted (Pred) values.** $k$NN (**a**, **c**, **e**) and DNN (**b**, **d**, **f**) models are developed for predicting logP (**a**, **b**), zeta potential (**c**, **d**) and cellular uptake (**e**, **f**). logP dataset contains 147 unique nanomaterials, including 123 GNPs, 12 PtNPs and 12 PdNPs. Zeta potential dataset contains 213 unique nanomaterials, including 148 GNPs, 6 AgNPs, 12 PtNPs, 12 PdNPs, 8 MONPs, 24 QDNPs, and 3 Dendrimers. Cellular uptake dataset contains 71 GNPs, which were tested against A549 cells. Root mean square error (RMSE) and correlation coefficients (R²) are also shown. RMSE_5CV and R²_5CV represent the RMSE and R² values for 5-fold cross validation, while RMSE_val and R²_val represent the values for external prediction. R²_CV and R²_val above 0.5 indicate high correlation between Exp and Pred values.

high predictability in recent modeling challenges in drug discovery[35,36], it performed differently in other studies[37,38]. Here, the lower predictability of DNN models is likely due to overfitting caused by too many neurons in the layers compared to the size of the input data. Both $k$NN (Fig. 5e) and DNN (Fig. 5f) cellular uptake models performed better (i.e., higher R² values) than the logP and zeta potential models.

The resulted models, especially the $k$NN models, can be used to predict new nanomaterials directly from their structures and assist rational nanomaterial design. Because the cellular uptake dataset consists of only one type of nanomaterial (GNP) so that the applicability of the resulted cellular uptake model can be

reliably applied to predict new GNPs. The logP and zeta potential datasets consist of various types of nanomaterials collected from different sources. The two models can be used to predict the properties of a wide range of nanomaterials. In addition, based on the same nanostructure annotation method, machine learning models were recently built to predict the inflammatory responses and cytotoxicity of various carbon nanoparticles[39]. Once a new nanomaterial is virtually designed using computer, its properties will be assessed using the developed models before chemical synthesis. This procedure will greatly save resources by prioritizing new nanomaterials with desired properties and/or cellular uptake potentials.

## Discussion

In summary, we constructed a universal nanomaterials database containing structure annotations suitable for direct computational modeling. The database currently contains 705 unique nanomaterials with multiple biological testing results. Structures of these nanomaterials were annotated and stored as PDB files that are retrievable from online portal. The new data being uploaded in the future will rapidly expand the database. We also developed several machine learning models using three property and bioactivity datasets in this database and showed the models had highly accurate predictability based on cross-validation and external validation results (i.e., $R^2 > 0.5$). The resulted models can be used to predict two critical properties and one bioactivity of new nanomaterials directly from their nanostructures. Some materials such as alloy nanomaterials[40], polymeric micelles[41], mesoporous nanomaterials[42], and metal-organic frameworks (MOFs)-based nanomaterials[43] were tentatively not included in the database because their nanostructures were poorly defined and the related publications currently lack quality control information on their synthesis. Other nanomaterials that were annotated still lack representative data in some target endpoints, for example, cellular uptake potentials. For the database to be more useful, there is still a need to generate more biological data of diverse nanomaterials.

## Methods

**Experimental data curation**. The database was compiled from in-house data (297 unique nanomaterials) and external data (408 unique nanomaterials). The in-house data were collected from our previously published studies (these references were provided in Supplementary References). The external data was collected by manual literature searching. This process resulted in more than 1000 papers with nanomaterial data for further examination. The data were included in the database with the following conditions satisfied: (1) the material (e.g., core atoms) and size information were provided in this paper; (2) the surface ligand structures can be annotated and transferred into SMILES; (3) the nano-bioactivity or physicochemical property data were provided with detailed experimental information. There are 69 publications that were identified to contain useful data by fulfilling all criterions (these references were provided in Supplementary References). Each publication was manually examined, and relevant structure information (e.g., core, size, and surface ligands), experimental data, and testing details were extracted from the corresponding papers. For raw data with size and shape information of a set of nanoparticles instead of a single molecular entity, the same core was set for all the nanoparticles in this data source. Data were also obtained directly from figures of published papers using PlotDigitizer. The surface ligand structures were converted to SMILES, which were shown in Supplementary Data.

**Nanostructure annotation**. For nanoparticles, the core atoms were first put together as a nano core based on the particle size information. Then the associated surface ligands were randomly placed on the core surface. For GNPs, AgNPs, PtNPs, PdNPs, MONPs, and QDs, the core of the corresponding nanostructure was generated by replicating the unit cell of the most thermodynamically stable crystal structures and then deleting atoms outside the input diameter data. The lattice parameters (e.g., unit cell lengths and angles) were obtained from the Materials Project (https://materialsproject.org/). For CNTs, the python toolkit scikit-nano (https://scikit-nano.org/) was applied to construct the carbon core (pristine CNTs). All the surface ligands were optimized before being grafted to the nano core. As for $C_{60}$, the SMILES obtained from the paper[44] were directly converted to PDB file. The PDB files of DnaNPs were either collected from the corresponding papers[45–48] or generated by the Legogen[49]. The PDB files of dendrimers were collected from corresponding papers[50–53]. For CPNTs, the nanostructures were generated by an in-house program written in C++[54]. In this procedure, the amino acids were firstly connected as various cyclic peptides through peptide bonds and then these cyclic peptides were stacked as CPNTs through H-bonds.

**Nanodescriptor generation**. At first, 126 tetrahedron fragments were generated for each nanostructure based on our previous study, which were calculated by combining the Delaunay tessellation and atom types[27]. In our previous study, the value of a nanodescriptor was calculated as the value of each tetrahedron electronegativity multiplied by its occurrences in the nanostructure. As described above, the range of nanomaterial size has a wide distribution in the current database. As a result, there will be a large difference of the tetrahedron occurrences between the large nanomaterials and small nanomaterials. In order to resolve this issue, property-based descriptors were also calculated in this study. The procedure can be described as follows: (1) The occurrence of each tetrahedron was converted to frequency (the occurrence of each tetrahedron divided by the total number of all the tetrahedrons in each nanostructure). (2) More atomic properties were introduced, which included the calculated radii ($R_{cal}$), the covalent radii ($R_{cov}$), the empirical radii ($R_{emp}$), the atom mass ($M$), the boiling point ($T_{bol}$), the density ($\rho$), the electron affinity ($E_{ea}$), the electronegativity ($\chi$), the heat of fusion ($\Delta H_{fus}$), the heat of vaporization ($\Delta H_{vap}$), the first ionization energy ($IE_1$), the second ionization energy ($IE_2$), the melting point ($T_{mel}$), the molar volume ($V_{mol}$), the specific heat ($Q$), the thermal conductivity ($\lambda$) and the valence ($q$). Then, these 17 property values of each tetrahedron were multiplied respectively by the corresponding tetrahedron frequency, as described in our previous study[27]. As a result, 17 descriptor matrices were generated that each descriptor matrix contained 126 individual descriptors (the tetrahedron fragments integrated with atomic properties). The calculated nanodescriptors for all nanomaterials are available from the web portal. After removing descriptors with limited information (e.g., with consistent values over all nanomaterials), total 680 nanodescriptors were used for modeling purpose. The nanostructure annotations and nanodescriptor generations were described in details in our previous papers[27,55].

**Computational modeling**. The datasets were split into training sets (80% of the original datasets) and test sets (20% of the original datasets). The training sets were used to build models, and the associated test sets were used to evaluate the developed models. The performance of each model was indicated by 5-fold cross validation within the training set and the external validation by predicting the test set. In this study, two different machine learning approaches were used to develop the computational models. The $k$-nearest neighbor ($k$NN) method used the weighted average of nearest neighbors as its prediction and employed a variable selection procedure to define neighbors[27,55], which was developed in-house (also available at http://chembench.mml.unc.edu/). The deep neural network (DNN) is a multi-layer feedforward neural network, which was implemented using Keras 2.2.4 (https://keras.io/) python deep learning library, with the TensorFlow backend. The DNN architecture used in this study included a sequence of five dense layers (three hidden layers), which were fully connected neural layers. Three hidden layers contained 512, 128, and 64 nodes, respectively. The relu was used as activation function to perform non-linear transformations. The dropout function, set as 0.2, was used to prevent overfitting of the resulting models. The rmsprop and mean squared error (MSE) were used as optimizer and loss function to compile the DNN model in this study. The learning rate was set as the default value of the rmsprop optimizer. Each DNN model was trained for 300 epochs.

## Data availability

All experimental data can be accessed from the Supplementary Data or from the Experimental data page of the web portal (http://www.pubvinas.com/).

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

## Acknowledgements

X.Y. and B.Y. were supported by the National Key R&D Program of China (2016YFA0203103), the National Natural Science Foundation of China (91543204 and 91643204), and the introduced innovative R&D team project under the "The Pearl River Talent Recruitment Program" of Guangdong Province (2019ZT08L387). W.W. and H.Z. were partially supported by the National Institute of Environmental Health Sciences (grant number R01ES031080, R15ES023148, and P30ES005022). We thank A. L. Chun of Science StoryLab for editorial service.

## Author contributions

H.Z. and B.Y. conceived and designed the study. H.Z. designed the project strategy. X.Y. curated the experimental data, constructed the web portal, simulated the virtual nano-materials, calculated nanodescriptors, built the models, and performed validation. A.S. designed, wrote and tested codes for constructing the virtual nanomaterials and guided several nanodescriptors calculation. W.W. helped analyze the results. X.Y., B.Y., and H.Z. wrote the paper. All authors have read and approved this paper.

## Competing interests

The authors declare no competing interests.
