## [Peer Review File · Nature Communications]

Reviewers' comments:

Reviewer #2 (Remarks to the Author):

The topic of this paper is very important. Databases containing information on the composition, structure, physicochemical properties, and biological effects has been listed as a key aim of several nano safety and nanomaterials projects and networks for more than a decade. The paper by Yan et al. is generally well written and would be of considerable interest to researchers working in nanotechnology. However, there are some issues that need to be resolved before the suitability for Nature Communications can be determined.

1. The reviewer tried to access the database via the specified link. The home page came up but attempts to access any data were unsuccessful, the pages took a very long time to load sometimes (I gave up before they loaded). Sometimes visualisation of the nanoparticles failed due to file incompatibilities or time outs. I downloaded several types of nanoparticle PDF files, these only contain structures not physicochemical or biological data. How are these endpoints accessed? Perhaps using MACCS structure-data (SD) file format would be better, as all the associated endpoint data could be included in the downloaded structure file. These issues made it very hard to assess the value of the database other than by the example given in the paper. The access latency needs to be addressed before the database is launched, is it behind a national firewall? Additionally, the home page is sparse with no explanatory information, help documents etc available. These would be very useful to those wanting to access the site.

2. As the authors note, there are a number of other nanosafety and nanomaterials databases around that contain similar information. The nanomaterials registry referred to in Table 1 seems to have disappeared and the link is broken. There are databases generated by the OECD and several EU COST Actions over the past decade. The main advantage of the authors' database is that data can be downloaded as PDB files that contain structures and physicochemical or biological data. Two new EU Horizon 2020 projects are also aiming to generate annotated databases for modelling. Have the authors done an exhaustive search of all relevant databases to ensure they are not missing some that are similar to the one they have developed?

3. Given that nanomaterials are presented as structures, and because of the reviewer's inability to download data from the authors' databases, has the inherent variability of nanoparticles been accounted for? Most nanoparticles are distributions of sizes and shapes rather than single molecular entities implied by the structures in the database. Does the database also make available raw data, controls, experimental protocols, experimental measurement error, replicates etc.?

4. Some of the plots of distributions of properties in the database (Figure 2) and models of cellular uptake (Figure 5) would be better as the log of the property, given that uptake varies by several orders of magnitude.

5. If the database contains valid data then it would be very useful for training machine learning models of nanomaterials properties. However, the lack of availability of the universal VINAS descriptor generation is a limitation. Could these descriptors be generated for each material in the database and added to it? Descriptors would presumably only need to be calculated once.

6. The details of how the machine learning models were calculated are insufficient. For example, the DNN method should list the transfer function, dropout rate etc.

7. Comparison of the machine learning models is best done using standard errors of estimation (training set) and prediction (test set) rather than r^2 values, which depend on the number of data points and parameters in the model (see e.g. Alexander et al. J Chem Inf Model 2015).

8. Supplementary Table S1 would be better in landscape format or as an Excel file so that it can be read more easily.

Reviewer #3 (Remarks to the Author):

I found this article to be in line with the content of Nature Communications publications on the topics of informatics and interoperable data, but had a few major problems with the tenets of the

paper and the presentation. Some fundamental revisions would be required to address these shortfalls in order to support publication. My position is that a universal database is not possible nor has it been presented here convincingly; however, other useful things have been presented, so that the messaging could be adjusted to allow publication of this work.

From my vantage point, I suggest a reframing of the contribution as follows:

1. The title includes the word “universal”, and the paper claims this as well. This reviewer strongly believes a universal nanomaterial database is not possible, realistic, or useful at this stage in the study of nanomaterials, and that pursuit of this goal diverts and dilutes investment in efforts that would advance nanoinformatics in a more fit-for-purpose manner. If, instead, the authors claimed to be proposing a universal 3-d annotation structure for nanomaterials, as opposed to a DB, that would be different. It seems that the novel contribution proposed here is in fact this 3-D annotation, which if adopted broadly, seems that it could in fact be very useful to any nanomaterial described in this way. If you change the claim of the paper substantially to center the development of this annotation, that would be a claim you could strongly support with your work here.

2. In addition to the annotation, a strong case study for its utility is included in the form of the example that is represented by the DB and the correlation exercise. The work to construct a sample DB of 705 NMs, and to link it to the three sample endpoints you have chosen, act as a proof of concept that strengthens the case for the viability of this 3-D annotation method. Perhaps modular progress could be made by others using the annotation method and building out more endpoints.

3. Another fundamental problem is that nanomaterial impacts are heavily dependent on the environment that surrounds them, but the paper does not mention or highlight the importance of these parameters. Ph, ionic strength, etc. are key determinants of the characteristics of a nanomaterial in any given situation. The DB of nanomaterial characteristics alone could not ever be used to predict the impacts – one of the impacts is even zeta potential, which is determined at least as much by the surrounding medium as by the initial state of the material. If these parameters are indeed included, I recommend they should be mentioned very clearly as necessary parts of the data requirements to support meaningful interpretation.

4. When introducing the quantitative study of structural similarity, the paper lacks an explanation of why structural similarity is of interest. It would be good to have a brief statement of the motivation for this, explaining that it is being done e.g. for correlation with impacts, and potentially to support read-across for regulation?

~Christine Hendren

Response to Editorial Comments.

Reviewer comments:

Reviewer #2: The topic of this paper is very important. Databases containing information on the composition, structure, physicochemical properties, and biological effects has been listed as a key aim of several nano safety and nanomaterials projects and networks for more than a decade. The paper by Yan et al. is generally well written and would be of considerable interest to researchers working in nanotechnology. However, there are some issues that need to be resolved before the suitability for Nature Communications can be determined.

1) The reviewer tried to access the database via the specified link. The home page came up but attempts to access any data were unsuccessful, the pages took a very long time to load sometimes (I gave up before they loaded). Sometimes visualisation of the nanoparticles failed due to file incompatibilities or time outs. I downloaded several types of nanoparticle PDF files, these only contain structures not physicochemical or biological data. How are these endpoints accessed? Perhaps using MACCS structure-data (SD) file format would be better, as all the associated endpoint data could be included in the downloaded structure file. These issues made it very hard to assess the value of the database other than by the example given in the paper. The access latency needs to be addressed before the database is launched, is it behind a national firewall? Additionally, the home page is sparse with no explanatory information, help documents etc available. These would be very useful to those wanting to access the site.

Response: We thank the reviewer for bringing up this important point. We redesigned the web portal and tested it in China, America, Europe and Australia by our colleague and collaborators for access speed. The feedbacks are very good. As the increase of speed is a constant process, we will continue improving it with technology and software updates. We will also provide online help and collect users' feedback and suggestions for further improvements. Currently:

1) We optimized the web server and tested the web portal using all popular browsers, including IE, Firefox, Chrome and *etc.* The web portal can be used in all these browsers.

2) The size of PDB files of individual can vary significantly due to the size of nanomaterials. The largest PDB file can be 90 MB. The open of large PDB file online using JSmol software may cause web time out if the internet speed is limited. To resolve this issue, we now provide the

screenshot of the nanostructures of large nanomaterials (*i.e.*, PDB file size > 20 MB) and the PDB file size information. And in the newly added “Tutorial” page, we also suggest users can download the PDB file first and open it using VMD software installed in local computers.

3) Following the reviewer’s suggestion, a general tutorial about how to use the web portal, as the “Tutorial” page, and contact information were added to the portal.

4) As requested by the reviewer, the biological data were classified by endpoints and added to the “Experimental Data” page of the portal. The biological data were also provided as “Supplemental file S1” of the manuscript.

2) As the authors note, there are a number of other nanosafety and nanomaterials databases around that contain similar information. The nanomaterials registry referred to in Table 1 seems to have disappeared and the link is broken. There are databases generated by the OECD and several EU COST Actions over the past decade. The main advantage of the authors' database is that data can be downloaded as PDB files that contain structures and physicochemical or biological data. Two new EU Horizon 2020 projects are also aiming to generate annotated databases for modelling. Have the authors done an exhaustive search of all relevant databases to ensure they are not missing some that are similar to the one they have developed?

Response: We thank the reviewer for pointing out this issue.

1) We rechecked all the links in Table 1 and updated the broken link of Nanomaterials Registry database, which has been migrated to nanoHUB (<https://nanohub.org/>).

2) We have searched all the relevant databases and added several new databases into Table 1 (Page 31).

3) We also added extra sentences to describe these new databases on Page 4 Line 73-75.

“Although, new file formats (*e.g.*, JSON¹⁷ and ISA-TAB-Nano²⁷) are also specially designed in several nanomaterial databases, such as eNanomapper and NANoREG, to store and manage the curated nanomaterial data. Nanomaterial entities (*e.g.*, composition, physicochemical properties and biological activities of the nanomaterials) in these databases exist as text outputs extracted directly from publications, ignoring nanostructure annotations that are critical for modeling studies.”

We are aware of the fact that several EU Horizon 2020 projects (*e.g.*, calibrate and NanoFASE) are making efforts to generate annotated nanomaterial databases. We are also in contact with

their scientists at the moment. The EU project is in a dynamic process, our database will be also in constant reconstruction and improvements. Therefore, in coming months, we will exchange with EU scientists about our mutual progresses. We will work with them to remove repetitive work and join our forces.

3) Given that nanomaterials are presented as structures, and because of the reviewer's inability to download data from the authors' databases, has the inherent variability of nanoparticles been accounted for? Most nanoparticles are distributions of sizes and shapes rather than single molecular entities implied by the structures in the database. Does the database also make available raw data, controls, experimental protocols, experimental measurement error, replicates etc.?

Response: The reviewer is absolutely right. We have to select high quality data with clearly chemistry information and characteristics. To further clarify this condition, we added the following text on Page 12 Line 268-269:

“Each publication was manually examined, and relevant structure information (*e.g.*, core, size and surface ligands), experimental data and testing details were extracted from the corresponding papers. For raw data with size and shape information of a set of nanoparticles instead of a single molecular entity, the same core was set for all the nanoparticles in this data source.”

The experimental data has been shared by the web portal and also included in the manuscript as Supplementary file S1. Besides, we added general experimental information along with the newly added “Experimental data” of the portal. To clarify this in the manuscript, we added the “Data availability” section in the end and the following text on Page 15 Line 328-329:

“All experimental data can be accessed from the Supplemental file S1 or from the “Experimental data” page of the web portal (<http://www.pubvinas.com/>).”

4) Some of plots of distributions of properties in the database (Figure 2) and models of cellular uptake (Figure 5) would be better as the log of the property, given that uptake varies by several orders of magnitude.

Response: We thank the reviewer for the suggestion. We revised the relevant figures (Figure 2h, Figure 5e and 5f) according to the reviewer’s suggestion and modified the sentence “In this database, cellular uptake capacity of all nanomaterials varied from $-1.87 \text{ g cell}^{-1}$ to 1.36 g cell^{-1} with a log₁₀-transformation (**Fig. 2h**).” on Page 7 Line 144.

Fig. 2 Overview of the nanomaterial database. a-h, Distributions of nanomaterials accounting to (a) nanomaterial type, (b) surface ligand number, (c) nanomaterial size, (d) logP, (e) zeta potential, (f) cell viability, (g) reactive oxidative stress (ROS) and (h) cellular uptake. Nanomaterials in the database show chemical, structural and biological diversity. The numbers in the brackets of (b) (c) represent the range of the surface ligand number and nanomaterial size, respectively.

Fig. 5 Correlations between experimental (Exp) and predicted (Pred) values for models developed based on *k*NN and DNN approach using datasets from the database. *k*NN (a, c, e) and DNN (b, d, f) models for logP (a, b), zeta potential (c, d) and cellular uptake (e, f). logP dataset contains 147 unique nanomaterials, including 123 GNPs, 12 PtNPs and 12 PdNPs. Zeta potential dataset contains 213 unique nanomaterials, including 148 GNPs, 6 AgNPs, 12 PtNPs, 12 PdNPs, 8 MONPs, 24 QDNPs and 3 Dendrimers. Cellular uptake dataset contains 71 GNPs, which were tested against A549 cells. Root mean square error (RMSE) and correlation coefficients (R^2) are also shown. RMSE_5CV and R^2_{5CV} represent the RMSE and R^2 values for 5-fold cross validation, while RMSE_val and R^2_{val} represent the values for external prediction. R^2_{CV} and R^2_{val} above 0.5 indicate high correlation between Exp and Pred values.

5) If the database contains valid data then it would be very useful for training machine learning models of nanomaterials properties. However, the lack of availability of the universal VINAS descriptor generation is a limitation. Could these descriptors be generated for each material in the database and added to it? Descriptors would presumably only need to be calculated once.

Response: We added all the calculated descriptors to the “Descriptors” page of the web portal and added sentences “As a result, 17 descriptor matrices were generated that each descriptor matrix contained 126 individual descriptors (the tetrahedron fragments integrated with atomic properties). The calculated nanodescriptors for all nanomaterials are available from the web portal.” on Page 14 Line 305-306. This information was also introduced in details in the “Tutorial” page of the web portal.

6) The details of how the machine learning models were calculated are insufficient. For example, the DNN method should list the transfer function, dropout rate etc.

Response: We thank the reviewer for this suggestion. We added the relevant information to Page 15 Line 322-326.

“The DNN architecture used in this study included a sequence of five dense layers (three hidden layers), which were fully connected neural layers. Three hidden layers contained 512, 128 and 64 nodes, respectively. The *relu* was used as activation function to perform non-linear transformations. The *dropout* function, set as 0.2, was used to prevent overfitting of the resulting models. The *rmsprop* and mean squared error (MSE) were used as optimizer and loss function to compile the DNN model in this study. The learning rate was set as the default value of the *rmsprop* optimizer. Each DNN model was trained for 300 epochs.”

7) Comparison of the machine learning models is best done using standard errors of estimation (training set) and prediction (test set) rather than r^2 values, which depend on the number of data points and parameters in the model (see e.g. Alexander et al. J Chem Inf Model 2015).

Response: We thank the reviewer for pointing out this issue. The reason that we used R^2 instead of standard errors to show the predictivity of resulted models is because of the different magnitudes of activity values in three datasets. To clarify this, we calculated both R^2 and standard errors, and added these values into discussion on Page 10 Line 215-219 and Figure 5. The new paragraph reads:

“The correlations between experimental and predicted values of the six resulting models based on *k*NN and DNN are shown in **Fig. 5**, which also includes the root mean square error (RMSE) and correlation coefficients (R^2). Overall, both R^2 and RMSE for 5-fold cross validation (R^2_{5CV} and $RMSE_{5CV}$) and external prediction (R^2_{val} and $RMSE_{val}$) are at the same order of magnitude, indicating the 5-fold cross-validation process and external prediction yielded similar results. All correlation coefficients (both R^2_{5CV} and R^2_{val}) were above 0.5, indicating that all six models successfully predicted the relationships between the annotated the nanostructures and target activities.”

8) Supplementary Table S1 would be better in landscape format or as an Excel file so that it can be read more easily.

Response: We uploaded Table S1 originally as an Excel file and it was transformed into the current format automatically by the system. We will work with editorial office to resolve this issue.

Reviewer #3: I found this article to be in line with the content of Nature Communications publications on the topics of informatics and interoperable data, but had a few major problems with the tenets of the paper and the presentation. Some fundamental revisions would be required to address these shortfalls in order to support publication. My position is that a universal database is not possible nor has it been presented here convincingly; however, other useful things have been presented, so that the messaging could be adjusted to allow publication of this work.

From my vantage point, I suggest a reframing of the contribution as follows:

1) The title includes the word “universal”, and the paper claims this as well. This reviewer strongly believes a universal nanomaterial database is not possible, realistic, or useful at this stage in the study of nanomaterials, and that pursuit of this goal diverts and dilutes investment in efforts that would advance nanoinformatics in a more fit-for-purpose manner. If, instead, the authors claimed to be proposing a universal 3-d annotation structure for nanomaterials, as opposed to a DB, that would be different. It seems that the novel contribution proposed here is in fact this 3-D annotation, which if adopted broadly, seems that it could in fact be very useful to any nanomaterial described in this way. If you change the claim of the paper substantially to

center the development of this annotation, that would be a claim you could strongly support with your work here.

Response: We agreed to the reviewer about the unsuitable use of the term “universal” and changed the title to “Construction of a web-based nanomaterial database by big data curation and modeling friendly nanostructure annotations”. The abstract was changed accordingly to reflect this change as:

“Modern nanotechnology research has generated numerous experimental data for various nanomaterials. However, the few nanomaterial databases available are not suitable for modeling studies due to the way they are curated. Here, we report the construction of a large nanomaterial database containing annotated nanostructures suited for modeling research. The database, which is publicly available through <http://www.pubvinas.com/>, contains 705 unique nanomaterials, including 414 gold nanoparticles, 17 silver nanoparticles, 12 platinum nanoparticles, 12 palladium nanoparticles, 80 carbon nanotubes, 48 buckminsterfullerenes, 34 quantum dots, 32 metal oxides nanoparticles, 21 DNA origami nanoparticles, 11 dendrimers and 24 cyclic peptide nanotubes. Each nanomaterial has up to six physicochemical properties and/or bioactivities, resulting in more than ten endpoints in the database. All the nanostructures are annotated and transformed into protein data bank files, which are downloadable by researchers worldwide. Furthermore, the nanostructure annotation procedure generates 2,142 nanodescriptors for all nanomaterials for machine learning modeling purposes, which are also available through the portal. This database provides a public resource for data-driven nanoinformatics modeling research aimed at rational nanomaterial design and other areas of modern computational nanotechnology.”

2) In addition to the annotation, a strong case study for its utility is included in the form of the example that is represented by the DB and the correlation exercise. The work to construct a sample DB of 705 NMs, and to link it to the three sample endpoints you have chosen, act as a proof of concept that strengthens the case for the viability of this 3-D annotation method. Perhaps modular progress could be made by others using the annotation method and building out more endpoints.

Response: In such a short period, it is not feasible to include another research group to perform modeling studies using this database and show results in this manuscript. However, as an extra

proof of concept, we have used this structure annotation method and developed predictive models of a set of carbon nanoparticles for their inflammatory responses and cytotoxicity (*Ecotoxicology and Environmental Safety*, **2020**, 191: 110216). We added this extra dataset of carbon nanoparticles into the portal. And the following text was added into Page 11 Line 233-235:

“Because the logP and zeta potential datasets consist of various types of nanomaterials collected from many different sources, these models can be used to predict the behavior of a wide range of nanomaterials. In addition, based on the same nanostructure annotation method, machine learning models were recently built to predict the inflammatory responses and cytotoxicity of various carbon nanoparticles⁴⁴.”

3) Another fundamental problem is that nanomaterial impacts are heavily dependent on the environment that surrounds them, but the paper does not mention or highlight the importance of these parameters. Ph, ionic strength, etc. are key determinants of the characteristics of a nanomaterial in any given situation. The DB of nanomaterial characteristics alone could not ever be used to predict the impacts – one of the impacts is even zeta potential, which is determined at least as much by the surrounding medium as by the initial state of the material. If these parameters are indeed included, I recommend they should be mentioned very clearly as necessary parts of the data requirements to support meaningful interpretation.

Response: We thank the reviewer for bringing up this important point. We added a new “Experimental data” page to provide all collected experimental data. Experimental conditions (*e.g.*, protocols, PH and *etc*) were provided with individual dataset on the “Experimental data” page of the portal. To clarify this in the manuscript, we added the “Data availability” section in the end and the following text on Page 15 Line 328-329:

“All experimental data can be accessed from the Supplemental file S1 or from the “Experimental data” page of the web portal (<http://www.pubvinas.com/>).”

4) When introducing the quantitative study of structural similarity, the paper lacks an explanation of why structural similarity is of interest. It would be good to have a brief statement of the motivation for this, explaining that it is being done *e.g.* for correlation with impacts, and potentially to support read-across for regulation?

Response: We thank the reviewer for this suggestion. We added the following text to clarify this on Page 8 Line 158-161.

“Chemical structure is the key to determine a molecule’s physicochemical properties and biological activities. The content that structurally similar molecules should exhibit similar bioactivities is the fundamental hypothesis of all quantitative structure-activity relationship (QSAR) and other relevant modeling studies^{33,34}. To quantitatively study the structural similarity among nanomaterials, we calculated the pairwise Euclidean distance for all nanomaterials.”